# Recovery of Waste with a High Iron Content in the Context of the Circular Economy

**DOI:** 10.3390/ma15144995

**Published:** 2022-07-18

**Authors:** Erika Ardelean, Ana Socalici, Oana Lupu, Diana Bistrian, Cristian Dobrescu, Nicolae Constantin

**Affiliations:** 1Faculty Engineering of Hunedoara, University Politehnica Timisoara, Revolutiei No. 5, 331128 Hunedoara, Romania; oana.lupu@student.upt.ro (O.L.); diana.bistrian@fih.upt.ro (D.B.); 2Materials Science and Engineering Faculty, University Politehnica of Bucharest, 060042 Bucharest, Romania; cristiandobrescu@yahoo.com (C.D.); nctin2014@yahoo.com (N.C.)

**Keywords:** ferrous waste, briquettes, recycling, recovery, circular economy

## Abstract

In order to apply the concepts that allow the transition from a linear to a circular economy, waste generators and/or processors must identify those variants that generate products that can be used as secondary raw materials, thus also respecting the actions governing sustainable development. This paper presents such a variant, the briquetting of waste with high iron content, waste generated on current flows in steel enterprises or deposited in industrial sites. The obtained briquettes are analyzed for chemical and mechanical characteristics so that can be used as secondary materials in the steel production. An optimization of the chemical composition using generic algorithms is also proposed in order to obtain the mechanical characteristics necessary for the proper handling of these products.

## 1. Introduction

In the context of sustainable development, the resource efficiency, reuse, and recycling of the steel in question of ferrous waste is very important for the steel industry. The efficiency and quality of raw materials and auxiliary materials is an integral part of the steelmaking process [1]. The transition from the linear economy to the circular economy requires action at all stages of the manufacture of ferrous alloys: from the extraction and transport of raw materials to the design of materials and products according to the new market requirements, to the production, distribution, and consumption of steel products, to their recycling and to the integrated management of the resulting waste, but also the recovery of the elements still useful from them. The steel industry must use all raw materials to their full potential, tending to zero waste in steelmaking [2,3]. Each co-product formed during the manufacture of steel, in the context of the circular economy, must be used in new products. This approach minimizes the amount of waste sent to landfills, reduces emissions, and preserves raw materials [2,4]. Co-products resulting from the steel industry (dusts, slurries, slag, milling) are recycled and transformed into new products for the industry. For example, 97% is recovered from the total amount of slag (BOF slag, EAF slag) produced (78% is used for cement manufacturing, 21% in road construction and the rest in agriculture as fertilizer) [3,5]. Powdery ferrous waste comes from the various operations of cleaning exhaust gases and wastewater, in dry or sludge form. Given the granulation of the dusts captured when cleaning gases, their storage raises special problems regarding the protection of the environment, so that almost all the steel companies that own such waste are looking for solutions to reintroduce them into the steel circuit [6,7,8].

By using simple processing operations of this small and powdery waste (agglomeration, pelleting, briquetting [9,10,11,12,13,14,15]), by-products can be obtained and can be used in the steel industry, when developing cast iron (agglomerates, pellets) or steel (reduced agglomerates, metallic pellets, lighters with adequate mechanical strength–Figure 1). The condition that these products can be recovered in the steel industry is not to pollute cast iron or steel with residual elements such as Zn, Cu, Sn, Cr, Ni.

If, through the repeated recycling of this waste, there is an increase in the content of residual elements, which would lead to the contamination of cast iron/steel, there must be applied methods of reducing the content of these elements [16,17,18] or this waste should be recovered by other industries and not by steel industry [13,19,20].

Powdery and small ferrous waste resulting from different phases of industrial processes (in most cases steel) represents an intrinsic value, which is determined by the ferrous content (chemically bound iron, sometimes metallic) that can properly replace the raw material, namely iron ore/cast iron/scrap iron, in steel processes [6,7,8]. In addition to iron, the main element, some wastes also have a high carbon content as well as basic or fluidifying component, which are useful in the recycling process. Recovery of these wastes and their reintroduction into the economic circuit is necessary both from an economic point of view, because it leads to the economy of raw materials and from an ecological point of view [11,12]. The share of the use of ferrous waste is expected to increase in the medium and long term as a secondary raw material in the load of steel-making aggregates (global ferrous scrap availability expected to increase to 1 bn tones in 2030 and 1.3 bn tones in 2050). Thus, large parts of the natural raw materials are replaced by waste or by-products, from the steel industry or from other industries, which have mineral content or similar properties [2,11]. 

By using by-products obtained from waste deposits or produced on current streams, both economic and ecological benefits are achieved, thus saving natural resources (iron ores). However, these by-products must meet conditions in terms of chemical composition and mechanical characteristics (during their use in processing aggregates).

## 2. Materials and Methods

### 2.1. Materials

The work presents the results obtained from the processing by briquetting of sludge (sintering sludge, sludge mill scale and ferrous sludge) resulting from the steel industry. From the point of view of the chemical and particle composition, the waste can be recovered by recycling, the choice of technology having to consider all its qualitative characteristics.

Powdery ferrous wastes were subjected to qualitative analysis, in order to determine the characteristics: chemical analyzes granulation and mineralogical and morphological analysis. 

In the sintering sludge, samples were taken from the sludge warehouse in Bataga (Hunedoara, Romania). The chemical composition of the samples taken is shown in Table 1, and among the physical characteristics, we mention: the bulk density is 1.18 kg/dm^3^ and the natural slope angle is 16°. The distribution of the grading classes has two well-defined areas that differ essentially by the form of the class distribution and is shown in Figure 2 and Figure 3. Figure 4 shows the morphological aspects corresponding to the sintering sludge samples. In the analyzed samples, the particles have different sizes and shapes, some slightly rounded, others polyhedral.

The sludge mill scale samples were taken from a steel plant, which were analyzed for the determination of the quality characteristics (the chemical composition was determined on the ARL 9400 XRF spectrometer—Thermo Fisher Scientific, Waltham, MA, USA). The data on the chemical analysis of the sludge mill scale under analysis are presented in Table 2 and the particle size distribution of the samples is shown in Figure 5. 

The ferrous sludge samples came from a steel plant producing raw iron powder. Moreover, for these, the chemical composition and the particle size composition were determined. In order to establish the particle size classes, the laboratory installation used is the vibrating sieving machine “Analysette 3” Fritsch (FRITSCH GmbH—Milling and Sizing, Idar-Oberstein, Germany) with a complete set of sieges (mesh sizes of the siege 450–25 μm). The slam sample was graded by refusal and the resulting particle size composition is shown in Figure 6. The chemical analysis of the samples taken is shown in Table 3 and the morphological analysis of ferrous sludge is shown in Figure 7.

### 2.2. Method

The previously analyzed iron wastes were briquetted in order to obtain secondary materials that can be used in steelmaking—the experiments taking place only in the laboratory phase. The briquetting process was chosen because it is a simple, reliable process, to which the force required for compaction can be adjusted. Moreover, the process can be easily automated so that a certain productivity can be achieved.

The lighter processing option was chosen without the addition of binders, the briquettes obtained, being intended for the steel industry, being used as a raw material for the elaboration of steels.

The experimental technology of processing briquettes consists of the following steps: -Preparation of powdery ferrous waste (ferrous sludge, sintering sludge, sludge mill scale) for the formation of raw batch;-Determination of the chemical and granulometric composition on the batch of materials subjected to processing;-Dosing the materials according to the established recipe, for a lighter weight of 1 kg;-Homogenization of the batch in the homogenization drum, the time for this operation being about 10 min;-Briquetting the batch with the help of hydraulic press;-Hardening of briquettes in the oven;-Determination of the quality characteristics (chemical composition and compressive strength) for each batch of briquettes.

The technological process is shown in Figure 8. Lighters were manufactured after 10 prescriptions, the composition of which is shown in Figure 9.

The test lighters were subjected to heat-resistance in an oven, in accordance with the diagram in Figure 10.

## 3. Results and Discussions

The iron content in the test lighters varied within 60–68.76%. Figure 11 shows the obtained lighters and their image (macro-structure) using the digital stereomicroscope model 520 Spin M-D.

For the assessment of the quality characteristics of resistance to handling and transport of lighters, three technological characteristics were determined by experiments: resistance to cracking, resistance to crushing, and crushing interval. For good handling and transport behavior, the literature [7,11] indicates for resistance to cracking R_f_ > 2 N/mm^2^, for resistance to crushing R_s_ > 3 N/mm^2^, and the crush range I_s_ = (0.2–0.35) R_f_.

The determination of the technological resistance characteristics for the test lighters was carried out using the universal compression/traction test machine LabTest 6.50 as shown in Figure 12.

For the resistance specifications the values obtained are:R_f_ = 4–9 N/mm^2^,R_s_ = 9–18 N/mm^2^,I_s_ ≥ 5 N/mm^2^.

To establish double correlation equations between the technology factors of resistance to compression of the test lighters, the data were processed in the Matlab calculation program, the optimization being made using genetic algorithms.

Genetic Algorithm (GA) is a search-based optimization technique based on the principles of Genetics and Natural Selection. It is frequently used to solve optimization problems, to find optimal or near-optimal solutions to difficult problems which otherwise would take a long time to solve. Genetic Algorithms is proven to have many parallel capabilities, optimizing both continuous and discrete functions and also multi-objective problems.

The following notations were taken into account:x_1_–Ferrous sludge, %x_2_–Sludge mill scale, %x_3_–Sintering sludge, %y_1_–R_f_–resistance to cracking, N/mm^2^y_2_–R_s_–resistance to crushing, N/mm^2^y_3_–I_s_ = R_s_ − R_f_-crush range, N/mm^2^

Using the response areas method [21,22,23] we will further determine the connection between the parameters of this process and its characteristic responses as areas in the multidimensional space of variables. In the experiments conducted according to this method, the independent variables are varied simultaneously, taking a limited number of values in the considered experimentation range, called levels. With the help of this method, although the three independent variables are varied simultaneously, their main and higher-order effects, as well as the interactions between them, can be determined separately. Changing the independent variables will automatically change the output data. The results thus obtained can be used to improve the performance of the studied process.

Considering the case of a process with three parameters x_1_, x_2_, x_3_ that can be varied within the limits, x_1a_ ≤ x_1_ ≤ x_1b_, x_2a_ ≤ x_2_ ≤ x_2b_ and respectively x_3a_ ≤ x_3_ ≤ x_3b_, the area in the plan of independent variables represents the experimental region, and the points of this surface, having different coordinates triplets of values (x_1i_, x_2i_, x_3i_) of the parameters, represent experimental points. The area on which the responses for each test point are located is the response surface of the considered characteristic of the process.

Applying the design method of experiments (Design of Experiments) [24,25,26] a scheme of the experiments necessary to be performed in the case of three parameters (factors) was generated, in which the factors take the extremes and central values in their fields of variation. This scheme of experiments is called the central experimental design or of the Box-Behnken type [24,25,26], and the factors considered for our experiment fall within the following limits: 20 ≤ x_1_ ≤ 100; 0 ≤ x_2_ ≤ 70; and 0 ≤ x_3_ ≤ 10.

The second-order models best approximate the response areas also called regression surfaces:(1)$$f\left(x_{1},{\;x}_{2},{\;x}_{3}\right)=\beta_{0}+\beta_{1}\cdot x_{1}+\beta_{2}\cdot x_{2}+\beta_{3}\cdot x_{3}+{\;\beta }_{4}\cdot x_{1}^{2}+{\;\beta }_{5}\cdot x_{2}^{2}+\beta_{6}\cdot x_{3}^{2}+\;\beta_{7}\cdot x_{1}\cdot x_{2}+\beta_{8}\cdot x_{1}\cdot x_{3}+\beta_{9}\cdot x_{2}\cdot x_{3}+\varepsilon$$

In matrix notation the relation (1) becomes: (2)$$\left[y\right]={\left[x\right]}^{T}\cdot \left[\beta \right]+\left[\varepsilon \right]$$
where [x] is the vector of their factors and contributions on the model;
(3)$${\left[x\right]}^{T}=\left[1{\;x}_{1}{\;x}_{2}{\;x}_{3}{\;x}_{1}^{2}{\;x}_{2}^{2}{\;x}_{3}^{2}{\;x}_{1}x_{2}{\;x}_{1}x_{3}{\;x}_{2}x_{3}\right]$$

[y] represents the vector of the observations of the N response in the experiments;

[ε] represents the vector of the measurement errors; 

[β] is the vector of the coefficients of the regression surface, to be determined.
(4)$$\left[\beta \right]=\left[\beta_{0}{\;\beta }_{1}{\;\beta }_{2}{\;\beta }_{3}{\;\beta }_{4}{\;\beta }_{5}{\;\beta }_{6}{\;\beta }_{7}{\;\beta }_{8}{\;\beta }_{9}\right]$$

For the determination of the β coefficients of the response surface with the help of experimental data, the most appropriate is the method of the smallest squares [21,23,25], which ensures a minimum dispersion of the coefficients determined.

For this purpose, the objective function of the form is considered:(5)$$F\left(\beta_{0},{\;\beta }_{1},\;\ldots ,{\;\beta }_{9}\right)=\sum_{i=1}^{N}{\left(y_{1}-f\left(x_{1},{\;x}_{2},{\;x}_{3}\right)\right)}^{2}$$

It follows that the determination of the coefficients of the response area is equivalent to the following problem of minimizing the objective function:(6)$$\underset{\beta }{\min}F(\beta_{0},{\;\beta }_{1},\;\ldots ,{\;\beta }_{9})=\sum_{i=1}^{N}{\left(y_{i}-f\left(x_{1},{\;x}_{2},x_{3}\right)\right)}^{2}$$
which leads to a system of algebraic equations of the type:(7)$$\frac{\partial F\left(\beta_{0},{\;\beta }_{1},\ldots ,{\;\beta }_{9}\right)}{\partial \beta_{0}}=0,\ldots ,\;\frac{\partial F\left(\beta_{0},{\;\beta }_{1},\ldots ,{\;\beta }_{9}\right)}{\partial \beta_{9}}=0$$

The coefficients of the response area that shape the process in the study are given by the expression:(8)$$\left[\beta \right]={\left({\left[x\right]}^{T}\cdot \left[x\right]\right)}^{-1}\cdot {\left[x\right]}^{T}\cdot \left[y\right]$$

In order to validate the regression model, it is necessary to calculate the correlation coefficient R^2^, which measures the “proximity” of the response surface from the experimental points and has the expression:(9)$$R^{2}=\frac{{\mathrm{SS}}_{R}}{{\mathrm{SS}}_{T}}$$
where: ${\mathrm{SS}}_{R}=\sum_{i=1}^{N}\left(f\left(x_{1},x_{2},{\;x}_{3}\right)-y_{i}\right){\;}^{2}$ is the sum of squares of errors in relation to experimental observations

${\mathrm{SS}}_{T}=\overset{N}{\underset{i=1}{\sum }}{\left(f\left(x_{1},x_{2},{\;x}_{3}\right)-\bar{y}\right)}^{2}$ measures the total variation of the N observations.
(10)$$\bar{y}=\frac{1}{N}\cdot \sum_{i=1}^{N}y_{i}$$

The problem of determining the optimal parameters that maximize the response surface was solved using genetic algorithms [27,28,29]
(11)$$\underset{\left(x_{1},x_{2},{\;x}_{3}\right){\;}^{\mathrm{optimum}}}{\max}f(x_{1},x_{2},{\;x}_{3})$$

The response surface that shapes the resistance to cracking y_1_ according to the parameters: x_1_–ferrous sludge; x_2_–sludge mill scale; x_3_–sintering sludge, has the expression:(12)$$y_{1}\left(x_{1},{\;x}_{2},{\;x}_{3}\right)=4.00+0.0219\cdot x_{1}+0.0393\cdot x_{2}-0.025\cdot x_{3}+0.000156\cdot x_{1}^{2}-0.000612\cdot x_{2}^{2}-\;0.0200\cdot x_{3}^{2}+0.00125\cdot x_{1}\cdot x_{3}+0.00429\cdot x_{2}\cdot x_{3}$$
having a correlation coefficient R^2^ = 88.62%.

The results of the optimization problem (11) are shown in Table 4. The response area (12) and related contour lines are shown in Figure 13 for the average value of parameter x_1_.

The response surface that shapes the crushing resistance y_2_ according to parameters: x_1_–ferrous sludge; x_2_–sludge mill scale; x_3_–sintering sludge, has the expression:(13)$$y_{2}\left(x_{1},{\;x}_{2},{\;x}_{3}\right)=10.00+0.0563\cdot x_{1}-0.0119\cdot x_{2}+0.117\cdot x_{3}-0.000104\cdot x_{1}^{2}-0.000340\cdot x_{2}^{2}-\;0.0567\cdot x_{3}^{2}+0.000179\cdot x_{1}\cdot x_{2}+0.00375\cdot x_{1}\cdot x_{3}+0.00857\cdot x_{2}\cdot x_{3}$$
having a correlation coefficient R^2^ = 88.50%.

The results of the optimization problem (11) are presented in Table 5. The response area (13) and related contour lines are shown in Figure 14 for the average value of parameter x_2_. 

The response surface that shapes the crushing range y_3_ according to parameters: x_1_–ferrous sludge; x_2_–sludge mill scale; x_3_–sintering sludge, has the expression: (14)$$y_{3}\left(x_{1},{\;x}_{2},{\;x}_{3}\right)=5.03+0.0531\cdot x_{1}-0.0298\cdot x_{2}+0.092\cdot x_{3}-0.000339\cdot x_{1}^{2}+0.000170\cdot x_{2}^{2}-\;0.0317\cdot x_{3}^{2}+0.00250\cdot x_{1}\cdot x_{3}+0.00429\cdot x_{2}\cdot x_{3}$$
having a correlation coefficient R^2^ = 76.51%.

The results of the optimization problem (11) are shown in Table 6. The response area (14) and contour lines are shown in Figure 15 for the average value of parameter x_3_.

From the correlation analysis shown in Figure 13 (level curves) it is noted that for sludge mill scale x_2_ = 33–37% and sintered sludge x_3_ = 3–7% (for x_2_ + x_3_ = 40%), for the crack resistance, the better accurate R_f ≥_ 6.5 N/mm^2^ are obtained.

By analyzing the correlation shown in Figure 14 it is noted that for ferrous sludge x_1_ = 55–62% and sintering sludge x_3_ = 3–10% (for x_1_ + x_3_ = 65%) very good values are obtained for shattering resistance, i.e., R_s_ ≥ 14 N/mm^2^.

A technological analysis of the correlation presented in Figure 15 allows the identification of optimal variation ranges for ferrous sludge x_1_ = 58–62% and sludge mill scale x_2_ = 33–37% (for x_1_ + x_2_ = 95%) so that very good values are obtained for shattering resistance, i.e., I_s_ ≥ 7.8 N/mm^2^.

## 4. Conclusions

In the “EU Action Plan for the Circular economy” presented in 2015, the European Commission supported the reasons why an appropriate, efficient, and rational use of resources, including through re-use/recycling/re-use of waste generated on current flows as secondary raw materials, can generate new and sustainable competitive advantages for the EU. Replacing the linear economy model and “closing the loop” of product life cycles through greater re-use and recycling–thus achieving a circular economy, can bring benefits for both the environment and the economy [30].

In the paper, the experiments carried out are aimed at identifying a technology and optimized alternatives whereby waste, generated on current flows or stored historically in areas where steel companies have operated, can be recovered by the industry where it originated, according to the concepts of the circular economy.

Experimental results lead to the conclusion that the waste analyzed (ferrous sludge, sludge mill scale, sintering sludge) can be processed by briquetting (with mechanical strength characteristics higher than the minimum values for this process), which allows the recovery of waste with large particle size variation limits (desirable below 2 mm).

The composition of the recipes can be determined based on the availability of small and powdery waste and the destination of the processed material: steelworks. By using a mathematical modeling with generic algorithms, optimal ranges of variation of the briquetting recipe components can be obtained (in the case of the three types of waste with the average composition presented in the paper: 33–37% sludge mill scale, 3–10% sintered sludge, 55–62% ferrous sludge).

Genetic algorithms are a modern tool used in artificial intelligence to look for a space for potential solutions to find a solution. They are faster and more efficient compared to traditional methods and help to find optimal solutions to difficult problems, while the traditional algorithm provides a step-by-step methodical procedure to solve a problem.

In order to obtain higher quality indicators for the products obtained (lighters), optimization of the kind described in the work may be used or some other changes to the raw batch recipes may be made, including by using a binder to eliminate hot hardening—in this case, much more rigorous control of the chemical composition is needed, especially from the point of view of the useful element: iron.

## Figures and Tables

**Figure 1 materials-15-04995-f001:**
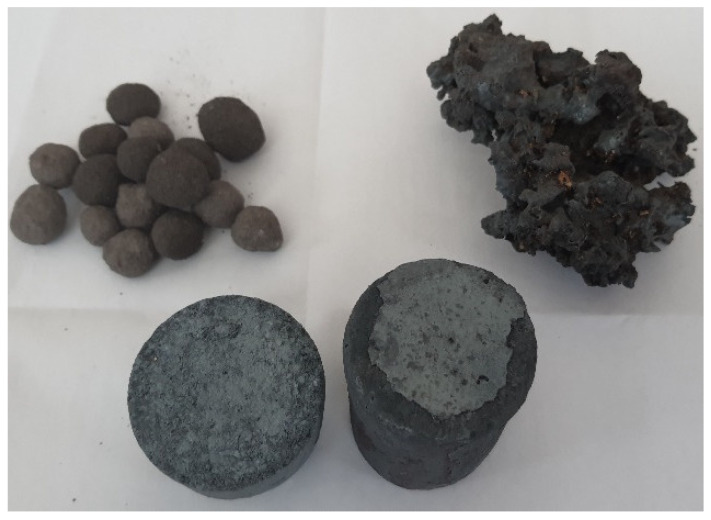
By-products made from waste with high iron content [7].

**Figure 2 materials-15-04995-f002:**
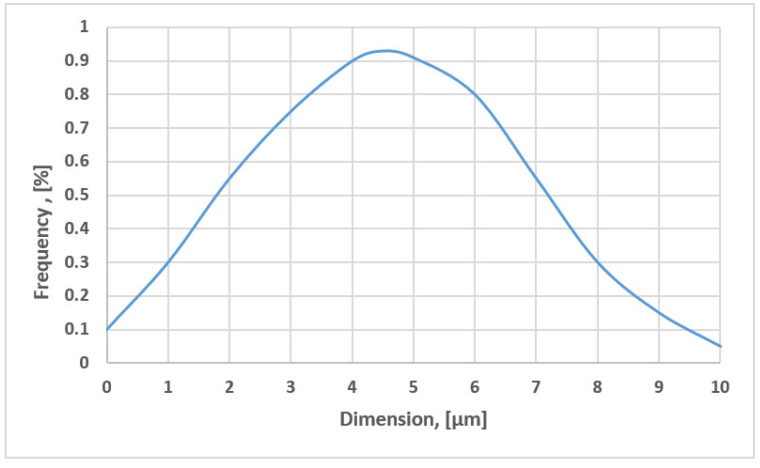
Distribution of particle size classes for sintering sludge, class 0–10 μm.

**Figure 3 materials-15-04995-f003:**
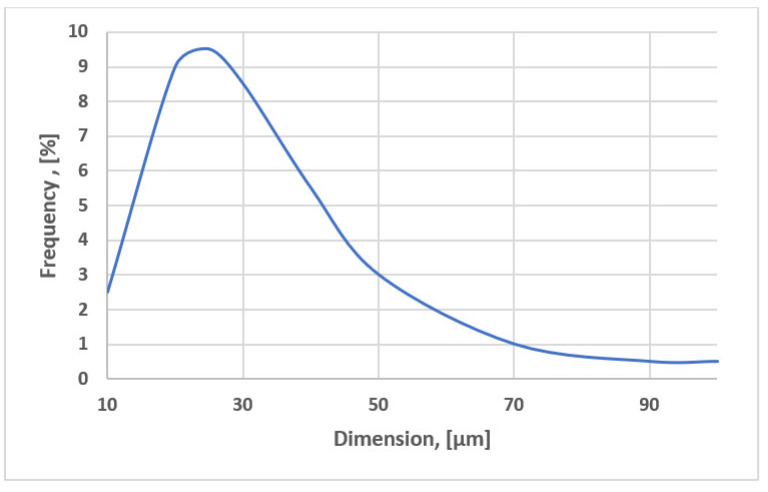
Distribution of the particle size classes for sintering sludge, class 10–100 μm.

**Figure 4 materials-15-04995-f004:**
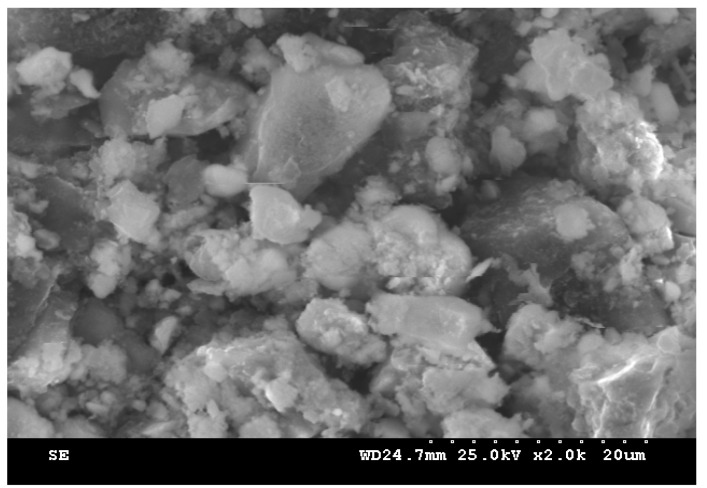
Image of sintering sludge, 4000×.

**Figure 5 materials-15-04995-f005:**
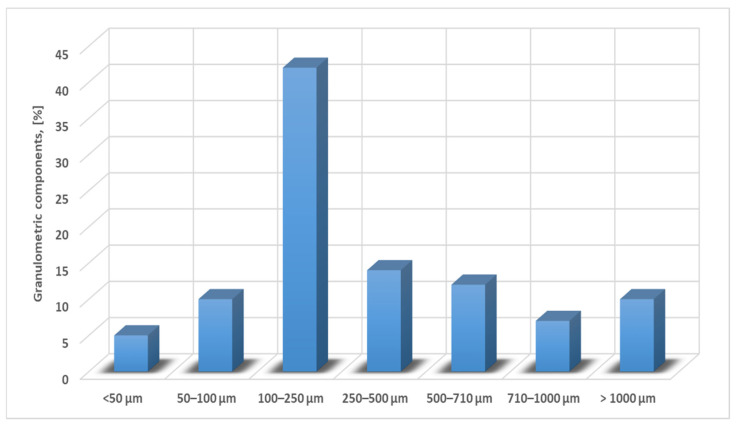
Particle size composition of sludge mill scale.

**Figure 6 materials-15-04995-f006:**
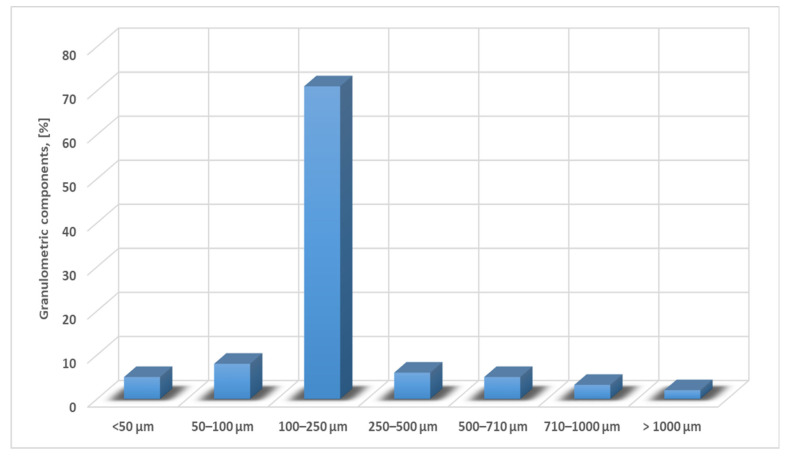
Particle size composition of ferrous sludge.

**Figure 7 materials-15-04995-f007:**
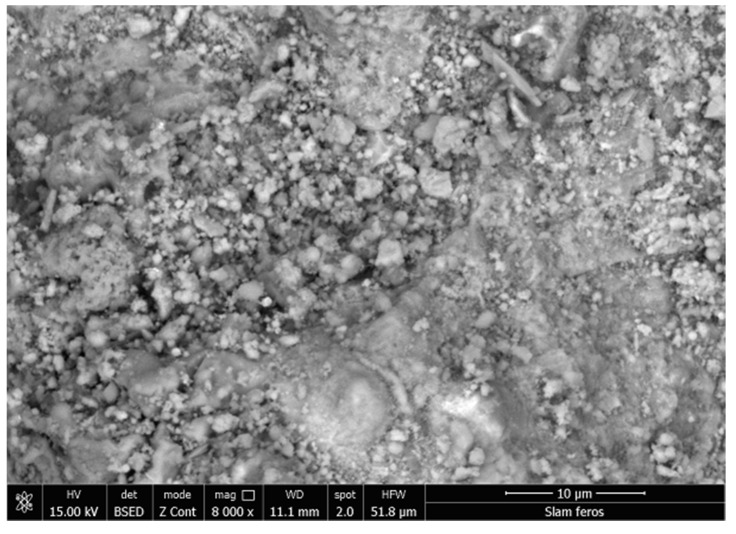
SEM images for ferrous sludge, 8000×.

**Figure 8 materials-15-04995-f008:**
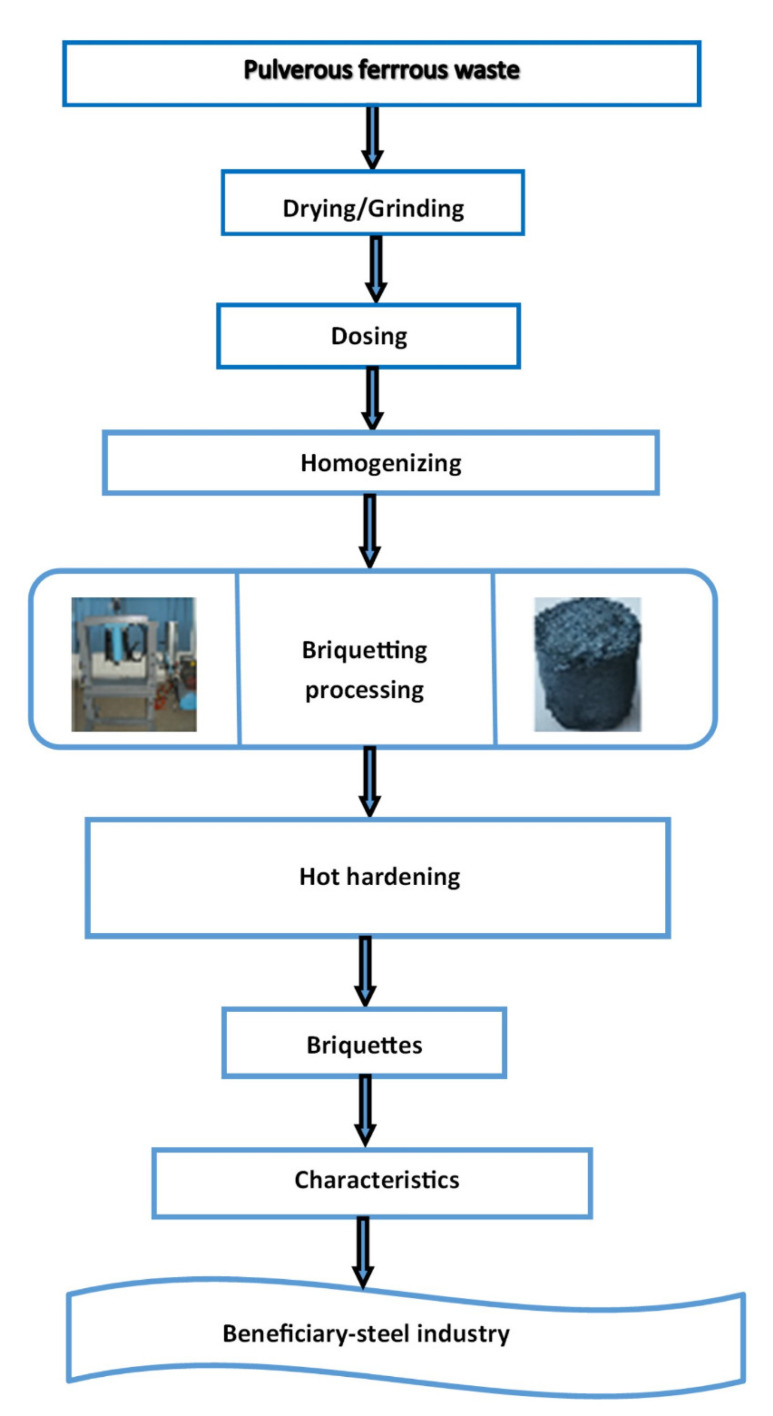
Technological stream of waste processing through briquetting.

**Figure 9 materials-15-04995-f009:**
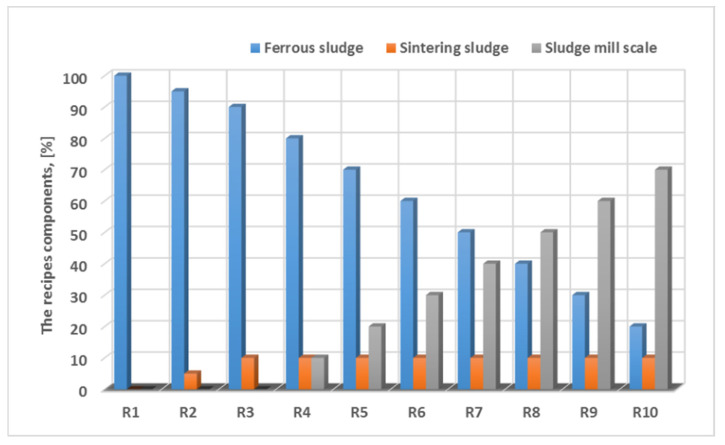
Composition of the experimental recipes.

**Figure 10 materials-15-04995-f010:**
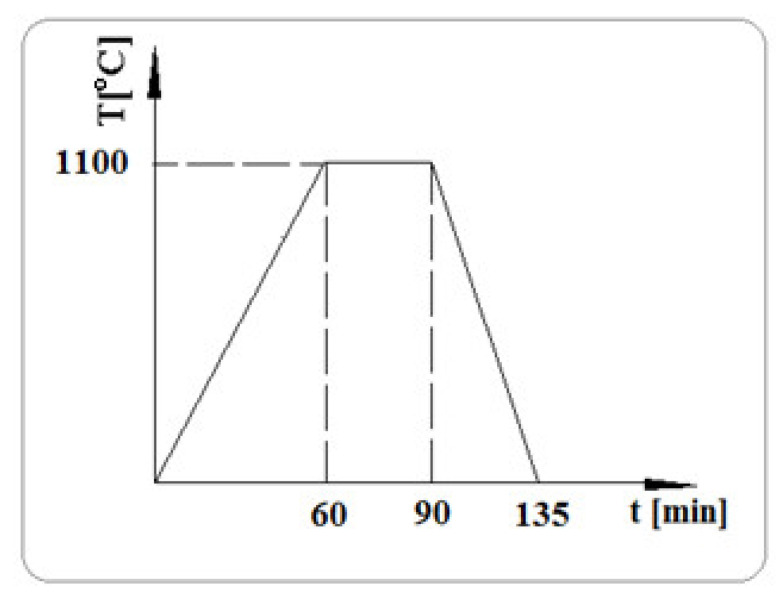
Lighter hardening diagram (T-hardening temperature, [°C]; t-hardening time, [min]).

**Figure 11 materials-15-04995-f011:**
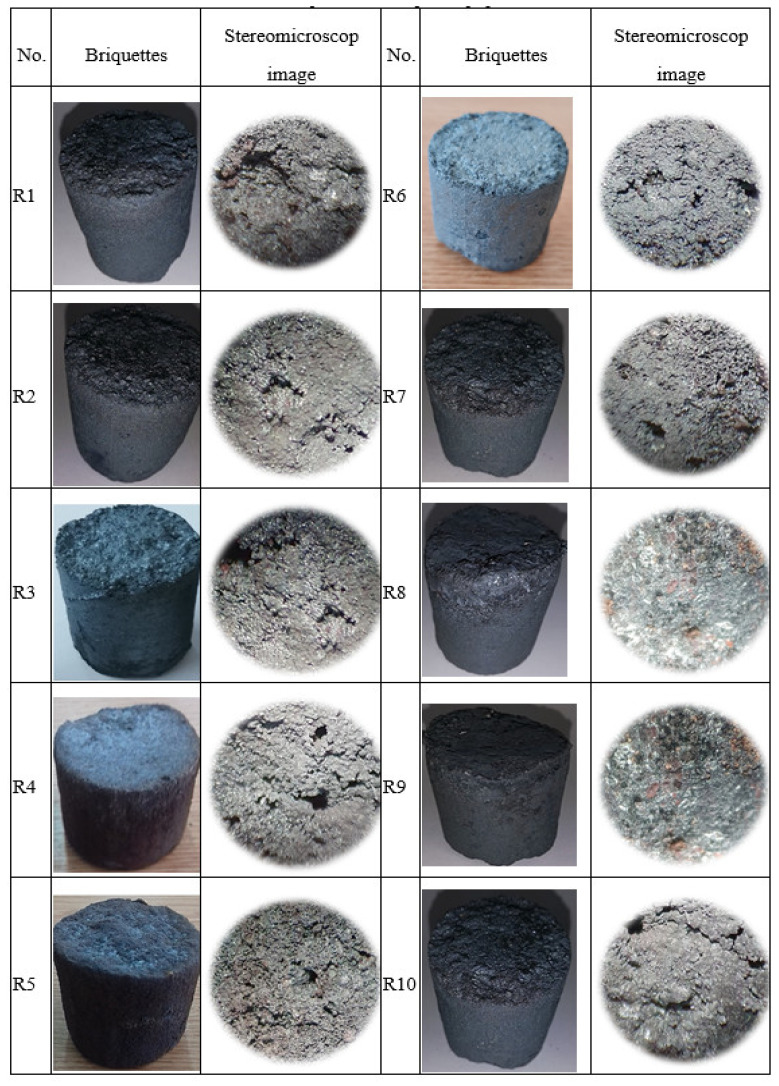
Experimental briquettes.

**Figure 12 materials-15-04995-f012:**
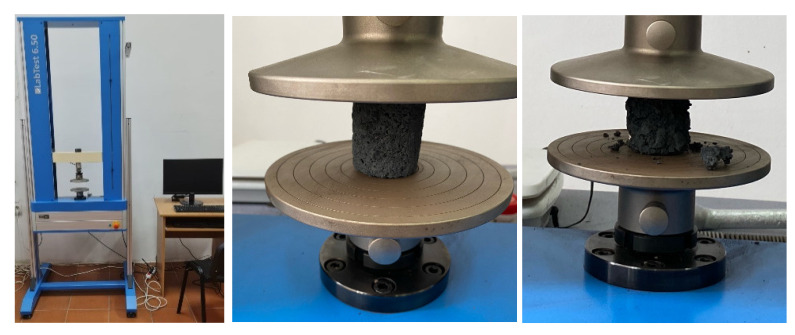
Determination of the briquette’s resistance characteristics.

**Figure 13 materials-15-04995-f013:**
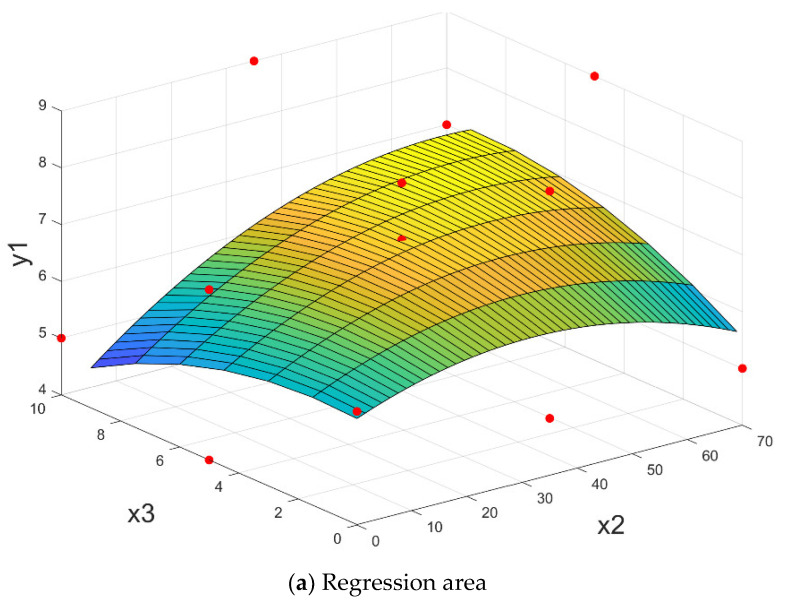
The briquettes resistance to cracking depending on the proportion of sludge mill scale and sintering sludge, for a medium value of ferrous sludge y_1_ = f (x_1med,_ x_2_, x_3_).

**Figure 14 materials-15-04995-f014:**
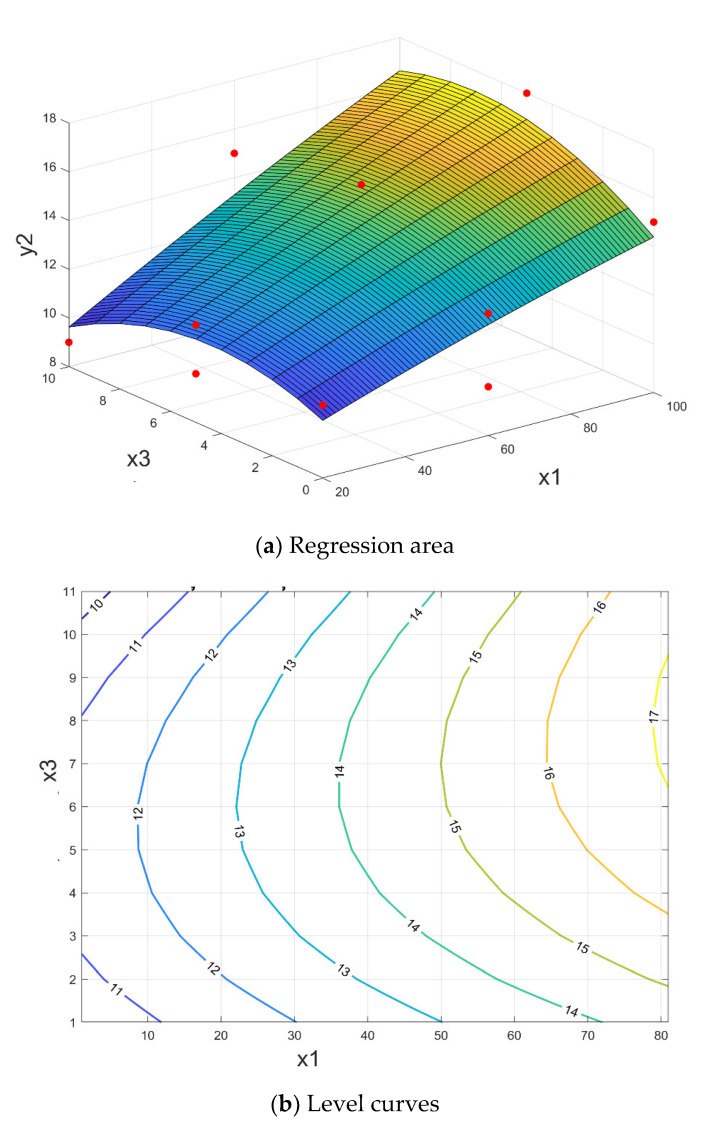
The briquettes resistance to crushing depending on the proportion of ferrous sludge and sintering sludge, for a medium value of sludge mill scale y_2_ = f (x_1,_ x_2 med_, x_3_).

**Figure 15 materials-15-04995-f015:**
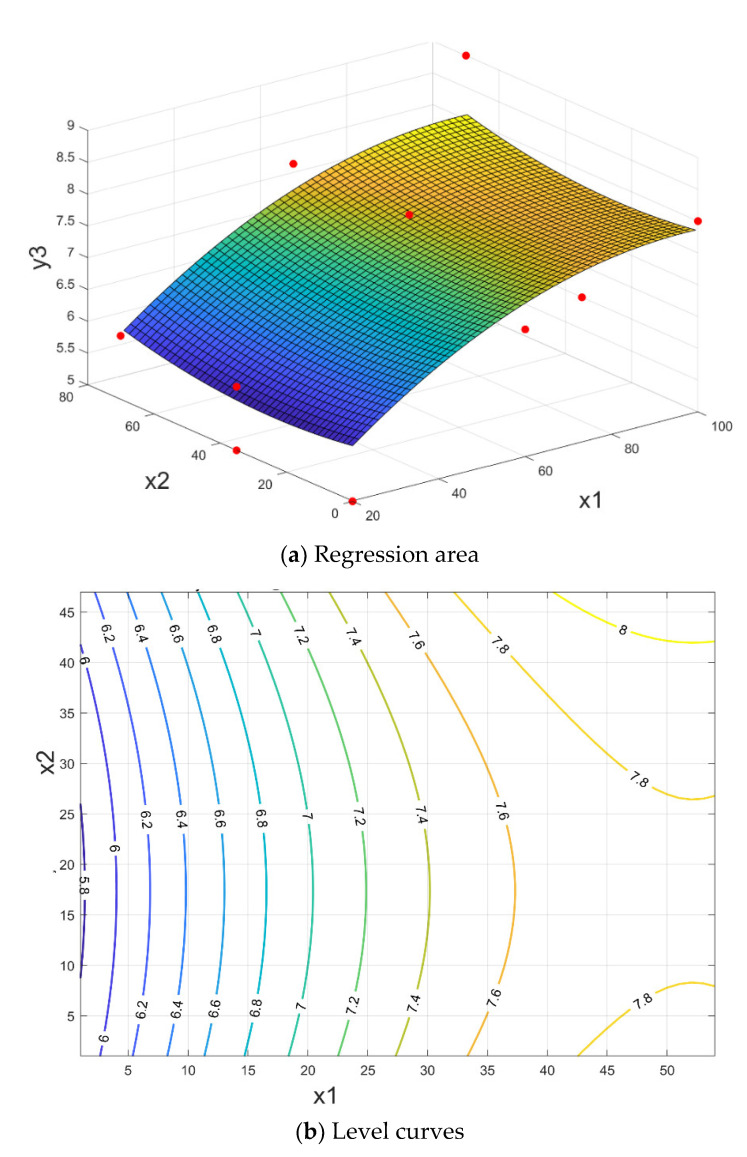
The briquettes crush range depending on the proportion of ferrous sludge and sludge mill scale, for a medium value of sintering sludge y_3_ = f (x_1,_ x_2_, x_3 med_).

**Table 1 materials-15-04995-t001:** Chemical composition of sintering sludge, [%].

Fe_tot_	FeO	Fe_2_O_3_	SiO_2_	Al_2_O_3_	CaO	MgO	MnO	S	P	C	CL *
31.43	8.51	35.44	9.90	9.88	9.91	2.55	0.85	1.53	0.17	21.15	1.19

* calcination losses.

**Table 2 materials-15-04995-t002:** Chemical composition of sludge mill scale, [%].

Fe	Ca	Si	Mn	Al	Mg	Cl	Na	Cr	Cu	Zn	Ni	Other Elements
89.97	2.14	1.91	1.97	0.87	0.60	0.43	0.38	0.28	0.23	0.11	0.08	1.03

**Table 3 materials-15-04995-t003:** Chemical composition of ferrous sludge, [%].

Al	Si	Cu	Cr	Mo	Mn	Ni	Pb	Fe	Sn	Sb	Zn	Other Elements
0.003	1.21	0.08	0.04	0.23	0.18	0.03	0.003	81.19	0.001	0.001	0.03	17.01

**Table 4 materials-15-04995-t004:** The results of the optimization problem for y_1._

Optimal Parameters (x_1_, x_2_, x_3_)	Maximum Value y_1_
(100.0000, 65.4843, 9.5232)	9.5130

**Table 5 materials-15-04995-t005:** The results of the optimization problem for y_2._

Optimal Parameters (x_1_, x_2_, x_3_)	Maximum Value y_2_
(100.0000, 70.000, 9.6293)	18.6088

**Table 6 materials-15-04995-t006:** The results of the optimization problem for y_3._

Optimal Parameters (x_1_, x_2_, x_3_)	Maximum Value y_3_
(100.0000, 70.000, 10.000)	8.9500

## Data Availability

Not applicable.

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
