# Peer review of "Recovery of Waste with a High Iron Content in the Context of the Circular Economy"

_materials, 2022, doi:10.3390/ma15144995_

Round 1
Reviewer 1 Report
The paper presents a solution for obtaining a valuable by-product for small and powdery waste holding a high iron content, which could be applied by the steel industry and its service providers.
The topic of the paper is interesting and well aligned to the aims and scope of the Journal and the considered special issue. However, the paper needs a considerable reworking to reach a sufficient quality for the publication.
Firstly, the analysis of the state of the art is quite superficial and does not duly consider the recent analyses which have been pursued in the steel sector on the topic of reuse and recycling of by-products in the steel sector. The authors can refer to the following exemplar papers:
https://doi.org/10.3390/met10030345
https://doi.org/10.1016/j.wasman.2021.04.008
https://doi.org/10.1177/0734242X21996552
Moreover, neither in the introduction nor in the following sections the innovative aspects of the proposed process clearly emerge. On the other hand, the introduction should be concluded with 1-2 paragraphs highlighting the main elements of novelty of the proposed solution, also based on the previous analysis of the state of the art, where the gaps to be filled should be more clearly identified. Furthermore, More details on these elements of novelty should be provided in Section 2.
Section 2 should be better organised, by splitting it into two subsections, namely materials and methods and the above-mentioned deeper details on the elements of novelty of the solutions should be provided in the second subsection.
In Section 3 the need and benefits of using Genetic Algorithms to identify the models parameters (see page 9) is not evident, as traditional techniques can be applied as well. Moreover, the paper misses a short section devoted to the discussion of overall meaning and the practical application of the achieved results. This could be even provided in the Conclusions section which, however, in its present form, is not effective in this sense. The Authors should elaborate more on this aspect.
Author Response
Thank you for the time allotted to the review and for the observations made. These are the changes made in the article:
- Reforming the Abstract – more clearly and to the point – paragraph 2,3,4.
- The analysis of the state of the art on the topic of reuse and recycling of by-products in the steel sector.
We have completed with the main ways of processing small and powdery waste taking into account recent publications – including the recommended ones, but only for siderurgic industry. We have introduced Figure 1 – with examples of such by-products from the processing of iron waste – own products. We also mentioned the presence of unwanted elements in the composition of this waste, which must be taken into account in the field of use of the products obtained – p.2, paragraph 1,2.
- The introduction should be concluded with 1-2 paragraphs highlighting the main elements of novelty of the proposed solution, also based on the previous analysis of the state of the art, where the gaps to be filled should be more clearly identified.
The solution presented – of briquetting the small and powdered waste is not a new procedure, but the novelty lies in the types of waste that make up the proposed recipes, the briquetting technology without the use of binder but with the hardening of the hot lighters, at the temperature of 1100oC– p.2, paragraph 4, p.5, paragraph 1.
- Section 2 should be better organised, by splitting it into two subsections.
We divided the subchapter into two parts: 2.1. Materials, 2.2. Method, and we have completed with the advantages of the briquetting process applied in the case of this waste – p.5, paragraph 1.
- In Section 3 the need and benefits of using Genetic Algorithms to identify the models parameters (see page 9) is not evident, as traditional techniques can be applied as well. Moreover, the paper misses a short section devoted to the discussion of overall meaning and the practical application of the achieved results. This could be even provided in the Conclusions section which, however, in its present form, is not effective in this sense.
We briefly presented the benefits of using generic algorithms both in Section 3 (p.9, paragraph 2) and in the Conclusions (p. 14, paragraph 7). It is also the application of the model used on the applied research presented, with the clear identification of the optimal intervals of variation of the recipe components for increasing the mechanical characteristics of the briquettes (p.14, paragraph 6).

Reviewer 2 Report
Dear authors,
Thanks for sharing your interesting manuscript. Your manuscript deals with the production of briquettes from mixtures of side-products of the steel making process. It describes the characterisation of these side-products and the relation between the feedstock mixture composition and two main mechanical properties of the produced briquettes. This technical characterisation and mechanical property analysis is well executed. My prime concerns with your manuscript is the suggested relationship with “circular economy” and the consequently vague title, abstract, introduction and discussion.
First of all, making something from production by products is not necessarily circular and does not necessarily fit in the concept of the circular economy as implied by the European Commission. And yes, I do agree that making something from side-products is useful and morally justified. But we cannot establish what the produced briquettes will be used for. Nor is it clear what will happen with these briquettes after their use and life-time. Based on the elemental composition of these side products and hence the briquettes, I suspect that future foundries will be extremely reluctant to use them as feedstock. So, the first recommendations that I urgently make to you is to specify which products could be made with your technique and to assess whether these products qualify the definitions of “recovery” or of “recycling” according to the definition in the European waste framework directive [98/2008/EU, article 3 points 15 and 17] and if it is eligible to account for recycling according to implementation decision EU 2019/665, annex III.
In case you can prove that these briquettes can be remelted and recycled in new alloys after use, than you can name this circular, if not it is just a form of “valorisation of side-products” or if you prefer “societal dumping of waste”. If so, you will need to rephrase the title, abstract, introduction and conclusion accordingly.
What I am lacking in your discussion is a reflection on the impact that your process could have to improve the material usage and material circularity of the European steel industry. Cooper et al. clearly described the limits of the American steel industry in circularity due to the accumulation of foreign elements in scrap steel [DOI: 10.1111/jiec.12971]. Can your process improve usage and “circularity”? In related articles the accumulation of foreign elements in “circular steel” is further elaborated [DOI: 10.1111/jiec.13246, DOI: 10.1016/j.resconrec.2021.105692]. If we consider those contributions, there is an urgent need to clean recirculated scrap steel, so how could the contaminated side products fit in a circular economy for steel? That is absolutely unclear to me.
The English language is in general clear and the article is well-written. What I, however, strongly dislike is the abstract that it filled with “circularity buzzwords” without providing information on what the article is truly about. The abstract should describe the content of this paper and not superfluous gobbledygook.
Small comment: Table 1: What is PC? Phosphor carbide?
Author Response
Thank you for the time allotted to the review and for the observations made. These are the changes made in the article:
- My prime concerns with your manuscript is the suggested relationship with “circular economy” and the consequently vague title, abstract, introduction and discussion. The first recommendations that I urgently make to you is to specify which products could be made with your technique and to assess whether these products qualify the definitions of “recovery” or of “recycling” according to the definition in the European waste framework directive [98/2008/EU, article 3 points 15 and 17]
By processing the iron-containing waste presented in the paper in the form of briquettes and the use of these briquettes in the steel industry (the products obtained are intended for steel elaboration – figure 8) we consider that we are part of the circular economy concepts. Based on your recommendations, we remain at the concept of waste recovery, according to Annex II of the above-mentioned document: R 4 Recycling/reclamation of metals and metal compounds.
- The abstract should describe the content of this paper and not superfluous gobbledygook.
Reforming the Abstract – more clearly and to the point – paragraph 2,3,4.
- We have completed with the main ways of processing small and powdery waste taking into account recent publications – including the recommended ones, but only for siderurgic industry. We have introduced Figure 1 – with examples of such by-products from the processing of iron waste – own products. We also mentioned the presence of unwanted elements in the composition of this waste, which must be taken into account in the field of use of the products obtained – p.2, paragraph 1,2.
- In case you can prove that these briquettes can be remelted and recycled in new alloys after use, than you can name this circular, if not it is just a form of “valorisation of side-products.
The briquettes obtained from the compaction of waste with iron content are used in the elaboration of the steel – under the conditions of rigorous control of the chemical composition that you mentioned yourself. The wastes used in our own experiments do not have high contents in these elements – we have not used steel dust that can, through a repeated recycling, bring these unwanted elements into steel in high quantity. There is an economic agent producing steel that currently uses briquettes with composition similar to those presented. The briquettes represent 15-20% of the total metal load of the elaboration furnace and, it was found on a concrete analysis of the chemical composition of the elaborated steel, that it did not increase the content in other elements beyond those provided in the elaboration standards. At this moment, we do not have the permission of the economic agent to disseminate the information.
- We completed the Conclusions with the benefits of using generic algorithms (p. 14, paragraph 7). It is also the application of the model used on the applied research presented, with the clear identification of the optimal intervals of variation of the recipe components for increasing the mechanical characteristics of the briquettes (p.14, paragraph 6).
- What is PC? Phosphor carbide?
Sorry. Material error. We made the necessary corrections: calcination losses – Table 1

Round 2
Reviewer 1 Report
The paper has been amended according to the suggestions provided in the first review round. The main weaknesses of the paper have been overcome in the second version. The paper is now suitable to publication.